# Styrene-Free Bio-Based Thermosetting Resins with Tunable Properties Starting from Vegetable Oils and Terpenes

**DOI:** 10.3390/polym14194185

**Published:** 2022-10-05

**Authors:** Fabio Bertini, Adriano Vignali, Marcello Marelli, Nicoletta Ravasio, Federica Zaccheria

**Affiliations:** 1Istituto di Scienze e Tecnologie Chimiche “Giulio Natta” (SCITEC), Consiglio Nazionale Delle Ricerche (CNR), Via Corti 12, 20133 Milano, Italy; 2Istituto di Scienze e Tecnologie Chimiche “Giulio Natta” (SCITEC), Consiglio Nazionale Delle Ricerche (CNR), Via Fantoli 16/15, 20138 Milano, Italy; 3Istituto di Scienze e Tecnologie Chimiche “Giulio Natta” (SCITEC), Consiglio Nazionale Delle Ricerche (CNR), Via Golgi 19, 20133 Milano, Italy

**Keywords:** biomaterials, vegetable oils, terpenes, biocomposites, thermosets

## Abstract

The substitution of fossil-based monomers in the thermosetting formulations is a fundamental issue to face the environmental concerns related to the use of traditional resins. In this paper, styrene-free thermosetting resins were prepared to start from vegetable oils with different compositions and unsaturation degrees, namely soybean, hempseed, and linseed oils. Using terpenic comonomers such as limonene and β-myrcene allows one to prepare thermosets avoiding the traditional fossil-based diluents such as styrene, thus obtaining an outstanding gain in terms of both environmental and safety concerns. Furthermore, the materials obtained reveal tunable physical properties upon the proper choice of the monomers, with glass transition temperature ranging from 40 to 80 °C and Young’s modulus ranging from 200 to 1800 MPa. The possibility of preparing composite materials starting from the resins prepared in this way and natural fibres has also been explored due to the potential applications of bio-based composites in several industrial sectors.

## 1. Introduction

The overall plastic production is ever increasing with a massive contribution coming from extra-European countries, with a coherent although limited increase in the production of the bio-based material, and the demand for thermosetting resins is even higher with respect to thermoplastic ones [1]. However, it is important to underline that the environmental concerns are expected to impose more constraints on thermosets due to their impossibility to be reprocessed [2]. In this respect, the recent studies have focused on the chemical recycling of these materials, which means the deconstruction of the polymer to re-obtain monomers or complex oligomers to be used for different purposes in a strategy that is different from the simple remoulding traditionally used in the case of thermoplastics [3,4]. This approach represents, of course, a viable route in order to capitalize on the starting carbon used for the plastic synthesis. Still, it requires significant money and energy-consuming processes in many cases.

Therefore, the use and exploitation of biomass-derived molecules as starting materials for thermosetting resins remains a desirable and convenient way in order to reduce the environmental impact of their end-of-life management, especially considering their very extensive use in electronic, aerospace, and automotive sectors and their common landfill and incineration end [5,6].

A collateral but not minor issue to be taken into account is the commercial and regulatory one, which more and more boosts the use of renewables in all the industrial and consumer areas, also resulting in the marketing and/or legislative needs of product eco-labels.

Among the bio-derived building blocks for the chemical industry, vegetable oils are definitely eligible for the preparation of bio-polymers, offering the great opportunity to tap into a pool of complex molecules in terms of carbon skeleton and various and reactive monomers in terms of double-bonds availability [7,8,9]. However, one of the main differences among vegetable oils of different origins resides in the profile of fatty acids linked with glycerol, which is in the different compositions in terms of both the length of the carbon chain and the number of double bonds present along the chain of the fatty acids (Figure 1). The latter is responsible for the oxidative stability of the oil and, therefore, for the oleochemicals obtained, both biodiesel and bio-lubricants, where high poly-unsaturation should be avoided [10,11].

In particular, the number of carbon–carbon double bonds could significantly impact the preparation of reticulated materials. The critical role of the oil unsaturation degree in the final properties of polymeric material has already been put in light in different formulations [12,13], and some of us have already reported the beneficial effect obtained in using a polyol prepared from highly monounsaturated triglycerides for the preparation of polyurethane foams [14]. In that case, a selective hydrogenation process allowed to reduce the unsaturation, maximize the oleic acid content, and obtain a regular content and position of the dangling chain obtained after polyol addition. This resulted in better performances of the final polymer in terms of flexibility.

On the contrary, a higher unsaturation degree could be an important requisite for fatty monomers intended to be used for highly crosslinked materials such as epoxy or epoxy-derived thermosetting resins [15].

A further step in obtaining bio-based carbon-containing materials resides in substituting the comonomer often used in addition to the major monomers [16,17]. The comonomer’s role, sometimes called the reactive diluent, is mainly to act as a linear chain extender, thus improving the overall polymer performance. On the other hand, this component is traditionally chosen among aromatic molecules such as styrene or divinylbenzene that, besides their fossil origin, also entail serious concerns related to their toxicity, as they are volatile organic compounds (VOC) and hazardous air pollutants (HAP) [18]. It is worth underlining that VOC emissions not only occur during all the phases of fabrication of polymers and composites but also after their cure; that is, during the lifetime of the material, they can be substantial.

Several attempts have been made in this direction by using bio-derived molecules as polyols and their derivatives [19,20]: rosin acids [21,22,23], lignin derivatives [24], and carbohydrates [25,26]. On the other hand, few attempts have been made to use terpenes as comonomers, namely eugenol [27,28].

In the main examples reported, the comonomers need chemical modifications to comply with the mechanical performances of the final materials, and the use of all bio-based components without important chemical modifications is still lacking.

Here, we wish to report our results in using oils with different unsaturation degrees as starting materials for the preparation of acrylated monomers and on their combination with terpenes as comonomer. Moreover, preparing composites with the same matrixes and natural fibre is presented as an example of a complete bio-based approach to preparing thermosetting materials by properly choosing the sources to obtain the desired performances.

The future researches aim to spread the use of bio-based composite materials in different industrial sectors, which shall minimize the pollution and improve the sustainability demands in the society [29,30].

## 2. Materials and Methods

### 2.1. Materials

Soybean oil epoxide (ESO) was kindly provided by GreenSwitch (Ferrandina, Italy); food-grade hempseed oil and linseed oil were used and purchased at food market; Amberlyst^®^ IR120 was purchased from Supelco (Sigma Aldrich, St. Louis, MO, USA); formic acid ACS puriss.; H_2_O_2_ solution (34.5–36.5% vol); acrylic acid anhydrous 99%; DABCO^®^ 32-LV (1,4-diazabicyclo[2.2.2]octane, Sigma Aldrich, St. Louis, MO, USA); Luperox^®^ (*tert*-Butyl peroxybenzoate, 98%,) and hydroquinone (>99.5%) were purchased from Sigma Aldrich (St. Louis, MO, USA). (±) Limonene (Li) (>95%), styrene (St) (99%), β-myrcene (My) (>70%) were purchased from TCI (Portland, OR, USA), α-Pinene (Pi) (>98%) was purchased from Fluka (Buchs, Switzerland). Some physical properties of the terpenes used as comonomers are reported in Appendix A.

### 2.2. Methods

In a typical epoxidation reaction, 100 mL of vegetable oil were charged in a three-necked flask equipped with a thermometer, a condenser, and a dropping funnel. In the flask were therefore added formic acid (13–16.5 g), toluene (50 mL), and Amberlyst^®^ (25 g, Sigma Aldrich, St. Louis, MO, USA). The suspension was heated at 60 °C, and H_2_O_2_ (90–113 g) was slowly added with the funnel by carefully controlling the temperature that increases during the addition of H_2_O_2_ and the in-situ formation of peroxide acid. At the end of H_2_O_2_ addition, the reaction was left to proceed for 5 h at 60 °C under magnetic stirring and, after that, cooled down at room temperature. Then, 25 mL of AcOEt were added, the suspension filtered with Buchner to separate Amberlyst^®^ (Sigma Aldrich, St. Louis, MO, USA), and then washed with warm water until reaching pH = 7. The organic phase was evaporated under reduced pressure, and the epoxide obtained was used for the following acrylation reaction [31].

The products obtained, namely the hempseed oil epoxide and the linseed oil epoxide, were analysed by ^1^HNMR and hereinafter called as EHO and ELO, respectively.

In a typical acrylation reaction, 50 mL of ESO, EHO, and ELO were charged with acrylic acid (16–24 g), hydroquinone (50 mg), and DABCO^®^ (30 μL, Sigma Aldrich, St. Louis, MO, USA) in a two-necked flask equipped with a condenser and a thermometer. The reaction mixture was warmed up to 90 °C, and the reaction was left to proceed under magnetic stirring for 11 h [32]. Therefore, the products obtained, namely acrylated epoxidised soybean oil (AESO), acrylated epoxidised hempseed oil (AEHO), and acrylated epoxidised linseed oil (AELO), were analysed by ^1^HNMR In a typical curing procedure; the desired amount of acrylated epoxidised oil (80% by weight) and the terpenic comonomer (20% by weight) were degassed under vacuum for 20 min and transferred into a silicon mould. Then, the proper amount of catalyst Luperox^®^ (0.0025% by weight, Sigma Aldrich, St. Louis, MO, USA) was added to the mould that was finally treated in a high-temperature furnace by applying the optimised temperature program (2 h at 140 °C, 2 h at 160 °C, and 4 h at 180 °C). The times and temperatures of the curing procedure were chosen based on the DSC analysis carried out on the final materials (*vide infra*). The optimised procedure allows one to obtain complete polymerisation.

The sample prepared with styrene as comonomer and used for reference purposes was treated with the following heating program due to the lower boiling point of styrene: 2 h at 120 °C, 2 h at 140 °C, and 4 h at 160 °C.

### 2.3. Thermoset Resins Characterization

The effectiveness of epoxidation and acrylation reactions was verified by ^1^HNMR analysis. The products obtained were analysed by a Bruker (Karlsruhe, Germany) Advance 400 spectrometer interfaced with a workstation equipped with a topspin software package. Typical ^1^HNMR spectra are reported in the Appendix A.

A Perkin Elmer (Waltham, MA, USA) TGA 7 thermogravimetric analyser investigated the volatility of comonomers. The analysis performed isothermal scans at 30 °C until the complete volatilization of monomers under nitrogen flow.

In order to verify the complete crosslinking of resins and to optimize the curing procedure, differential scanning calorimetric (DSC) analysis was performed using a Perkin Elmer (Waltham, MA, USA) DSC 8000. The samples were heated from 0 to 240 °C at a rate of 20 °C/min under nitrogen flow (Appendix A).

A Perkin Elmer (Waltham, MA, USA) TGA 7 thermogravimetric analyser studied the thermal stability of resins. The analysis at a heating rate of 20 °C/min from 50 to 750 °C in nitrogen atmosphere.

The tensile mechanical tests were carried out on dumbbell specimens (overall length 75 mm, gauge length 25 mm, a width of narrow section 4 mm) obtained by cutting the cured resins. The tests were conducted using a Zwick-Roell (Ulm, Germany) Z010 dynamometer with a load cell of 2.5 kN operating at a crosshead speed of 5 mm/min until break. The Young’s modulus (*E*), the maximum strength (*σ**_max_*), and the elongation at break (*ε_b_*) were averaged on at least five tests per sample and reported with their standard deviations.

Dynamic mechanical analysis (DMA) was performed using a TA Instruments (New Castle, DE, USA) DMA Q800 equipped with a single cantilever configuration, at a constant oscillation frequency of 1 Hz and a strain amplitude of 10 μm, in the temperature range from −40 to 160 °C with a heating rate of 3 °C/min.

Scanning electron microscopy (SEM) of the composites was performed by a Philips (Amsterdam, Netherlands) XL-30 ESEM instrument operating in low vacuum mode—0.8 torr, 12kV at a working distance of 11.7 mm. Samples were cryo-fractured and the exposed section analysed without surface treatment, simply fixing it onto a SEM stub.

## 3. Results and Discussion

### 3.1. Homopolymers with Acrylated Vegetable Oils

The direct epoxidation of carbon–carbon double bonds is one of the most used ways to easily functionalize the carbon chains due to the possibility of adding a wide variety of reactive groups to the oxiranic ring. Furthermore, this first step allows one to open the way to different kinds of monomers upon the desired final products [33].

Therefore, we designed the preparation of the fatty monomer according to a two-step sequence based on epoxidation followed by the addition of acrylic acid (Figure 1), already explored by some other authors [32].

Of course, the use of vegetable oils with different compositions in terms of unsaturation degree gives rise to a coherent extent in functionalization due to the proper choice of the reagent ratio depending on the number of double bonds. In Table 1, the composition of the three oils used as starting materials is reported [10,34], making evident the great difference in acrylic group number being able to be inserted.

A smart indicator of the unsaturation degree of vegetable oils is represented by the iodine value (IV), which accounts for the number of double bonds based on the mass of iodine needed for their oxidation: the higher the iodine value, the greater the number of double bonds. Values reported in the last column of Table 1 give a clear idea of the strong difference in terms of the unsaturation of the oils chosen as substrates.

The epoxydised and subsequently acrylated triglycerides obtained according to the reactions described in Figure 1 were therefore used as monomers for the following curing step by varying the temperature program of the curing procedure, taking care into account that the starting temperature has to be lower than the boiling point of the comonomer when present. This means that in the case of the resins prepared with styrene as the comonomer, a starting temperature of 120 °C has to be maintained, whereas terpenes can start at 140 °C.

The effect of the parent vegetable oil’s unsaturation degree on the obtained resins’ mechanical properties was evaluated by performing tensile tests by uni-axial stretching until the break. In Figure 2, the stress-strain curves of AESO, AEHO, and AELO are reported.

From the data reported, it is evident that the degree of unsaturation of the oil strongly influences the final mechanical properties of the resins: by comparing the homopolymers based on soybean, hempseed, and linseed oils, it was observed that the higher the unsaturation degree, the higher the material performance, that is, the combination of high strength and high stiffness. This aspect is attractive for producing polymer materials that are used in various applications such as structures and interior and exterior components in the civil industry (e.g., wind turbine blades) and transport industry (e.g., automotive and naval ones).

### 3.2. Heteropolymers with Acrylated Vegetale Oils and Terpenes

As already mentioned in the introduction, the addition of a comonomer also strongly affects the mechanical properties of the materials playing a role in the reticulation process, resulting in different strengths and stiffness of the resins. Therefore, the present work explores the possibility of substituting the traditional use of styrene- or fossil-based reactive diluents with natural products. In particular, the possibility of using non-modified terpenes was envisaged. Limonene, β-myrcene and α-pinene were chosen due to their non-oxygenated composition on the one hand and their different structure and C=C double bond availability (see Figure 3).

The effect of their addition to the monomers could help shed light on the importance of the open-chain vs. cyclic structure and/or the double bond availability to obtain an effective reticulation.

Figure 4 reports the mechanical properties of the samples, namely Young’s modulus (*E*) and maximum tensile strength (*σ**_max_*), confirming the aforementioned tensile behaviour as a function of the unsaturation degree of the starting oils. Moreover, the elongations at break (*ε_b_*) of samples are listed in Appendix A and present values range from 2% to 6%.

It is worth noting that the resins with β-myrcene led to the best results in terms of mechanical resistance regardless of the oil used. On the other hand, the presence of α-pinene gave the poorest performance, while the addition of limonene brought intermediate properties. This result highlights that in the case of β-myrcene, the presence of multiple double bonds with higher reactivity promotes a better co-polymerization and crosslinking. Moreover, comparing the different co-polymers, the performances in tensile properties were further increased in the presence of a more unsaturated oil such as linseed oil. In fact, the resins based on AELO, as previously mentioned, are the most crosslinked and show higher rigidity. It is worth underlining that using this kind of terpenes in preparing epoxy resins is still largely unexplored. Thus, despite several studies on the use of limonene and limonene oxide for the synthesis of non-isocyanate polyurethanes [35], polycarbonates [36], and innovative bio-based thermosets [37,38] or on the use of myrcene for thermoplastic or elastomers [39,40], very few papers rely on the use of limonene or myrcene with acrylated or epoxidised vegetable oils. The co-polymerisation of AESO in combination with myrcene is reported for 3D application [41]. The use of limonene and myrcene with tung oil has been proposed [42], obtaining materials endowed with quite poor *E* values.

An interesting point arises from data reported in Figure 4 by comparing the tensile properties of the above-described resins with the ones prepared to start from AESO and styrene. *E* and *σ**_max_* values obtained with this reference material clearly show higher mechanical properties with respect to the other resins based on AESO. On the other hand, looking at AELO and AELO-My resins, it appears evident that materials with similar properties or even more performance can be obtained thanks to the appropriate choice of the starting oil and the comonomer.

This effectively witnesses the great opportunity offered by using vegetable oils as starting materials for the preparation of monomers: the starting composition allows one to tune the crosslinking and in turn the tensile behaviour of the final product while giving the chance to avoid the use of styrene. As already underlined in the introduction, the substitution of styrene is desirable, as it is classified as a VOC for low boiling point and high saturated vapour pressure. Even worse, styrene is not environmentally friendly as an HAP and is harmful to human health [43]. The comonomers proposed here are not only naturally derived reagents but also reveal much lower volatility, as shown by the TGA measurements (Appendix A). The isothermal tests at 30 °C highlighted longer times of volatility for the three bio-based comonomers compared to styrene. Indeed, the latter took less than 50 min to reach the whole mass loss, while in the case of limonene, the time to obtain the same result was nearly 200 min. The slower volatility of the terpenic comonomers and their lower toxicity, compared to styrene, ensure a greater/high safety for the operator.

The evaluation of the bio-based content of the prepared materials gives a quick idea of the effect obtained in the substitution of styrene with terpenes. A product’s bio-based content is defined as the “amount of bio-based carbon in the material or product as a percent of the weight (mass) of the total organic carbon in the product” [44]. The starting oil (i.e., soybean oil as a benchmark) is obtained from renewable materials and contains ~77% carbon. After epoxidation and acrylation the total carbon content goes down to ~64% and the bio-based carbon to ~50%. A typical resin preparation mixture is based on 80% by weight of AESO and 20% by weight of comonomer. Therefore comparing the resin obtained with AESO and limonene (100% bio-based) and the one obtained with AESO and styrene (0% bio-based), the calculation of the bio-based content according to Pan et al. [45] is the following:
(1)$$\begin{array}{c} \mathrm{Bio-Based}\;\mathrm{Content}\;\%\;\mathrm{AESO-Li}\;=\\ \;\frac{\left(50\%\times 0.80\right)+\left(88\%\times 0.20\right)}{\left(64\%\times 0.80\right)+\left(88\%\times 0.20\right)}=83.7\% \end{array}$$
(2)$$\begin{array}{c} \mathrm{Bio-Based}\;\mathrm{Content}\;\%\;\mathrm{AESO-St}\;=\\ \;\frac{\left(50\%\times 0.80\right)}{\left(64\%\times 0.80\right)+\left(92.3\%\times 0.20\right)}=57.0\% \end{array}$$

It is worth noting that the development of fermentation processes for producing acrylic acid from biomass has seen a great boost in the last years and could reach mature technologies due to the great importance of this intermediate for the chemical industry [46]. Using bio-based acrylic acid would further increase the amount of bio-based carbon in the final resin, eventually reaching 100%.

The thermal stability and the degradation mechanism of cured resins were evaluated by TGA, monitoring samples’ mass loss with increasing temperature.

Appendix A shows the thermal decomposition curves of all resins. The temperature at which a mass loss of 5% occurs (*T_d,_*_5%_) was chosen as the one corresponding to the beginning of the degradation of the samples, and the *T_d,5%_* values are reported in Table 2. In general, both homopolymers and copolymers present good thermal stability and very similar thermal behaviour with a decomposition stage that occurs in the temperature range between 300 and 500 °C. In detail, the resins based on AESO and AEHO show high starting decomposition temperatures with values between 330 and 340 °C, while in the case of AELO-based samples, *T_d,_*_5%_ values are slightly lower (310–326 °C).

The dynamic mechanical properties of resins were measured through temperature scans between −40 and 160 °C. Figure 5a presents the storage modulus (*E*′) of some selected specimens as function of temperature.

The *E*′ values at −30 °C and 100 °C, respectively, relating to glassy and rubbery regions, are listed in Table 2. Since the degree of unsaturation of linseed oil is the highest among the investigated oils, the AELO-based resins exhibit larger storage moduli than other samples as previously well-observed by tensile mechanical tests. Moreover, *E*′ is further improved by the addition of β-myrcene, indicating that this comonomer acts as a stiffening agent for a crosslinked network of AELO. Comparing the homopolymers or the resins with the same comonomer, the same trend resulting from mechanical characterization was observed: the AEHO-based materials show lower storage moduli than AELO-based ones (especially in the glassy region), whereas *E*′ values are higher as compared to AESO-based resins. The damping factors, determined as the ratio of the loss modulus to storage modulus (tan *δ*
*= E*″/*E*′) of the resins, are depicted in Figure 5b. The α transition, assimilated to the glass transition, was attributed to the maximum temperature peak of tan *δ* and was reported as *T_g_* in Table 2. The resins based on linseed oil show the highest *T_g_* among the analysed specimens with values above 70 °C. AEHO-based resins exhibit *T_g_* around 60 °C, while the lowest values (*T_g_* between 40–50 °C) were observed in the case of AESO-based samples. Moreover, the tan *δ* peaks of the resins based on AESO and AEHO (Appendix A) are broader than those of AELO-based samples, highlighting a more heterogeneity of crosslinked systems [19,47,48,49]. In particular, this behaviour indicates that the length of the chain segments between the crosslinks is larger in AESO-based and AEHO-based resins than to AELO-based materials [50]. This result suggests that the resins based on linseed oil present a more rigid structure and higher crosslinking density associated with a reduced free volume. The reference AESO-St material shows dynamic mechanical properties (in terms of storage modulus and *T_g_*) definitely superior to AESO-based resins but comparable and lower than those based on hempseed and linseed oils, respectively. Once again, as well as observed from tensile tests, it emerges that it is feasible to prepare high-performance thermosets without using monomers from fossil origin.

### 3.3. Composites with Natural Fibers

An important part of thermosetting resins is employed in preparing composites for the automotive and naval industry or the construction sector [51,52]. Therefore, a step forward in the preparation of fully bio-based materials could reside in using natural fibres to prepare composites starting from the above-described resins. In the present work, two different composites were prepared to start, respectively, from AEHO and hemp fibres and AEHO, limonene, and hemp fibres (Figure 6). First, composite laminate was produced alternating in a mould with seven layers of woven hemp fabric wetted with the investigated resins by the hand-layup technique at room temperature. Afterward, the mould was placed in a furnace by applying the curing procedure previously described.

SEM micrographs of fractured surfaces, depicted in Figure 7, show good fibre impregnation in the resin. The absence of a gap between the matrix phase and the fibre highlights that the matrix fully encloses the fibre, proving that bonding at the matrix-fibre interface is strong. Furthermore, the adhesion of the matrix on fibres along with good wetting fibre was obtained without any fibres surface modification. This suggests that the high interfacial strength observed might ensure satisfactory mechanical properties of the composite [53,54]. The possibility of using chemically unmodified fibres is not trivial, as this aspect is reported to be a critical point [55]. Some interesting results on effective matrix–fibre interactions have been recently reported in the preparation of thermosets based on epoxidized vegetable oils and reinforced with flax fibres [56].

Actually, an optimum in fibre–matrix adhesion is generally reached by means of various compatibilization approaches that can range from matrix-oriented methods, where the polymer matrix is modified by chemical or physical procedures [57] or combinations of both to methods for fibre modification (chemical, physical, physicochemical, biological, or combinations of methods) or the use of additives [58]. Our preliminary results are very encouraging in this respect with a view to future and thorough research about the development of bio-based composites.

## 4. Conclusions

The design of the above-described materials entails several strength points from the sustainability point of view related to both the safeness of the products and the bio-derived origin.

First, the use of vegetable oils and terpenes as main sources allow one to obtain 84% of the overall carbon from biomass resin (even more if the composite is considered). This makes the polymer deserve the label of bio-based material.

Moreover, using terpenes such as β-myrcene or limonene instead of styrene avoids using a toxic reagent, which is already demonstrated to be volatile and to remain partially unreacted. The proper choice of the starting oil is an important tool for preparing materials that have as much performant as the styrene-containing ones. The linseed oil-derived materials reveal mechanical properties comparable or even superior to those measured with the soybean oil and styrene resin.

A further step forward was completed in designing composites that are almost fully bio-based, tapping from a unique productive chain, as is shown in the case of the hempseed oil and hemp fibre example reported in the present paper.

## Data Availability

Data is contained within the article or Appendix A.

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
