# Peer review of "Styrene-Free Bio-Based Thermosetting Resins with Tunable Properties Starting from Vegetable Oils and Terpenes"

_polymers, 2022, doi:10.3390/polym14194185_

Round 1

Reviewer 1 Report

I hope that you are ok,

Please find the below comments for the manuscript entitled (Styrene-free biobased thermosetting resins with tunable prop-2 erties). I suggest publication after major consideration that I have mentioned in the next lines.

Kind regards

Comments considerations:

1.     The abstract need to support by more results.

2.     The introduction needs to provide sufficient background and include all relevant references.

3.     What are the primary and secondary aims and consequences of the study?

4.     What are the advantages of the Styrene-free biobased thermosetting resins?

5.     All methods need to be supported by more references.

6.     Figures quality is very low and it is impossible to evaluate the results from stretched graphs and sometimes labels not visible at all. Use a homogeneous format, please.

7.     Put items and titles in the results and discussion section showing the characteristics of the prepared material.

8.     Please use SI units. All abbreviations and acronyms should be defined at first mention as  AESO, AEHO and AELO. 

9.     Has this material been tested in terms of real application?

10.                        I understand it is not easy to write a scientific paper in a second language; however, authors are responsible for writing concise and accurate texts. It is important because 1) poor writing does not help deliver a take-home message explicitly as well as can damage your valuable work. 2) not all readers are native English speakers, who can easily catch what authors would intend even if some sentences are grammatically broken. Please have a professional language proofreading service on this manuscript.

11.                        Title of the manuscript should express the main goal of the conducted research by a short and brief sentence describing the contents of the manuscript, and avoiding abbreviations and informal English language

12.                        Adjust chemical functional groups and units.

Author Response

The authors warmly thank the referee 1 for the suggestions. The manuscripts has been modified as follows according to the comments

Reviewer 1

Please find the below comments for the manuscript entitled (Styrene-free biobased thermosetting resins with tunable properties). I suggest publication after major consideration that I have mentioned in the next lines.

Point 1: The abstract need to support by more results.

The abstract has been modified by inserting the mechanical and thermal properties of the materials obtained.

Point 2: The introduction needs to provide sufficient background and include all relevant references.

In the introduction some recent references have been added. In particular refs [15], [17], [29], [30].

Point 3: What are the primary and secondary aims and consequences of the study?

In the abstract two sentences underlining the aims and the potential consequences of the study in terms of application have been added.

Point 4: What are the advantages of the Styrene-free biobased thermosetting resins?

The importance of substituting styrene is related to its volatility and toxicity and has been evidenced in the introduction section as follows:

“On the other hand, this component is traditionally chosen among aromatic molecules such as styrene or divinylbenzene that, besides their fossil origin, also entail serious concerns related to their toxicity, being volatile organic compounds (VOC) and hazardous air pollutants (HAP) [16]. It is worth underlining that VOC emissions not only occur during all the phases of fabrication of polymers and composites but also after their cure, that is, during the lifetime of the material, they can be substantial”

Point 5: All methods need to be supported by more references.

References relying on epoxidation and acrylation procedures have been added in the methods section, refs [31], [32].

Point 6: Figures quality is very low and it is impossible to evaluate the results from stretched graphs and sometimes labels not visible at all. Use a homogeneous format, please.

As requested the quality of the figures has been improved

Point 7: Put items and titles in the results and discussion section showing the characteristics of the prepared material.

The result and discussion section has been divided in three parts with the proper subtitles relying on the main families of materials prepared: homopolymers, heteropolymers and composites.

Point 8: Please use SI units. All abbreviations and acronyms should be defined at first mention as  AESO, AEHO and AELO. 

Acronyms AESO, AEHO and AELO have been defined at first mention in the Materials and methods section.

Point 9: Has this material been tested in terms of real application?

The prepared materials have not been tested in terms of real applications yet.

Point 10: I understand it is not easy to write a scientific paper in a second language; however, authors are responsible for writing concise and accurate texts. It is important because 1) poor writing does not help deliver a take-home message explicitly as well as can damage your valuable work. 2) not all readers are native English speakers, who can easily catch what authors would intend even if some sentences are grammatically broken. Please have a professional language proofreading service on this manuscript.

We did some language corrections and we warmly thank the editorial office for the fruitful work in this respect.

Point 11: Title of the manuscript should express the main goal of the conducted research by a short and brief sentence describing the contents of the manuscript, and avoiding abbreviations and informal English language

We expanded the title underlining the main biomass derived components of the materials proposed in the manuscript.

Point 12: Adjust chemical functional groups and units.

The units have been corrected and added when lacking in the figures.

Reviewer 2 Report

The paper presents an interesting approach based on the Styrene-free biobased thermosetting resins with tunable properties. However, the innovation of the current research work should be further highlighted and emphasized. At the same time, the authors should consider the following comments to greatly improve the quality of the paper.

1. In the abstract, add a final statement that highlights the importance of this research and its possible potentials. Also, introduce the problem in the initial lines of the abstract.

2. The introduction needs to be improved by relating to the mechanics of the studied materials and their mechanical characteristics. The references to be included are: 10.1177/0021998318790093, 10.1016/j.polymertesting.2017.09.009, 10.1016/j.compstruct.2021.114698, 10.1177/0731684417727143, 10.1002/app.46770, 10.1016/j.porgcoat.2022.107015.

3. Kindly add a table that describes the main physical and chemical properties of the raw materials used in this study.

4. Were the methods described by the authors come in accordance with a certain standard or do they follow previous procedures?

5. For the tensile tests, what was the geometry of the samples and what was the type of gribbers?

6. How many tensile samples were tested per configuration? Based on which standard, did you follow the mechanical testing?

7. What was the accelerating voltage and working depth in the SEM testing of the samples?

8. In Figure 4, why there is no error bar for each data point? The variance and standard deviation in each case is elmentary in describing the performance of these materials.

9. The conclusion needs to be modified to summarize the research outcomes in short statements with clear observations.

Author Response

The authors warmly thank the referee 2 for the suggestions. The manuscripts has been modified as follows according to the comments

Reviewer 2

The paper presents an interesting approach based on the Styrene-free biobased thermosetting resins with tunable properties. However, the innovation of the current research work should be further highlighted and emphasized. At the same time, the authors should consider the following comments to greatly improve the quality of the paper.

Point 1: In the abstract, add a final statement that highlights the importance of this research and its possible potentials. Also, introduce the problem in the initial lines of the abstract.

The abstract has been implemented according to referee's suggestions by adding the following sentences at the beginning and at the end:

“The substitution of fossil-based monomers in the thermosetting formulations is a fundamental issue to face the environmental concerns related to the use of traditional resins.”

“The possibility of  preparing composite materials starting from the resins prepared in this way and natural fibers has also been explored due to the potential applications of biobased composites in several industrial sectors”

Point 2: The introduction needs to be improved by relating to the mechanics of the studied materials and their mechanical characteristics. The references to be included are: 10.1177/0021998318790093, 10.1016/j.polymertesting.2017.09.009, 10.1016/j.compstruct.2021.114698, 10.1177/0731684417727143, 10.1002/app.46770, 10.1016/j.porgcoat.2022.107015.

The most fitting reference has been added in the introduction section.

Point 3: Kindly add a table that describes the main physical and chemical properties of the raw materials used in this study.

A Table describing the physical properties of the terpenes has been added in the Supplementary Materials (Table S1), whereas vegetable oils have been already described in Table 1.

Point 4: Were the methods described by the authors come in accordance with a certain standard or do they follow previous procedures?

We did not follow any standard, the tensile tests were performed using internal procedures (described in the Methods section).

Point 5: For the tensile tests, what was the geometry of the samples and what was the type of gribbers?

The tensile mechanical tests were carried out on dumbbell specimens (overall length 75 mm, gauge length 25 mm, width of narrow section 4 mm) using Zwick-Roell pneumatic grips, as described in the Methods section.

Point 6: How many tensile samples were tested per configuration? Based on which standard, did you follow the mechanical testing?

The mechanical properties were averaged on at least five tests per sample. As abovementioned the mechanical tests followed internal procedures reported in the manuscript.

Point 7: What was the accelerating voltage and working depth in the SEM testing of the samples?

Scanning Electron Microscopy (SEM) of the composites were performed by a Philips XL-30 ESEM instrument operating in low vacuum mode – 0.8 torr, 12kV at a working distance of 11.7 mm. Details have been added in the Methods section.

Point 8: In Figure 4, why there is no error bar for each data point? The variance and standard deviation in each case is elmentary in describing the performance of these materials.

The average value and the standard deviation of the tensile parameters, i.e. Young's modulus, maximum tensile strength and elongation at break, are reported for all investigated materials in the Supporting Materials (Table S2).

Point 9: The conclusion needs to be modified to summarize the research outcomes in short statements with clear observations.

Conclusions have been modified according to referee’s suggestions in order to make simpler and more vivid  the final results.